# Effects of Dietary Aged Maize with Oxidized Fish Oil on Growth Performance, Antioxidant Capacity and Intestinal Health in Weaned Piglets

**DOI:** 10.3390/ani9090624

**Published:** 2019-08-29

**Authors:** Bin Luo, Daiwen Chen, Gang Tian, Ping Zheng, Jie Yu, Jun He, Xiangbin Mao, Yuheng Luo, Junqiu Luo, Zhiqing Huang, Bing Yu

**Affiliations:** Institute of Animal Nutrition, Sichuan Agricultural University, Chengdu 611130, China

**Keywords:** aged maize, oxidized fish oil, growth performance, nutrient digestibility, antioxidant capacity, intestinal health, weaning piglets

## Abstract

**Simple Summary:**

In China, large quantities of maize are stored in grain depots for two years or more to mitigate the risk of natural disasters impacting feed supplies. However, it is unknown whether the use of long-term stored maize in diets will impair growth performance of piglets, and whether additional dietary oxidants would further exacerbate the effects. This study investigates the effects of dietary aged maize with the supplementation of different levels of oxidized fish oil on growth performance, nutrient digestibility, serum antioxidant activity and gut health in piglets and tries to provide a theoretical foundation for the better use of aged maize in swine production. The results of this study showed that aged maize had no significant effect on growth performance, diarrhea and nutrient digestibility of the piglets, but it did reduce serum antioxidant capacity. When oxidized fish oil was added, aged maize reduced serum antioxidant capacity further, inhibited the expressions of genes related to intestinal nutrient transport, promoted intestinal inflammation, and also reduced the apparent total tract digestibility (ATTD) of nutrients, increased diarrhea and finally reduced the growth performance of piglets. Thus, the use of aged maize in the diet of the piglets may be not feasible, especially when other oxidation-inducing factors existed, which would exacerbate the negative effects of the aged maize.

**Abstract:**

This study aimed to determine the effects of dietary aged maize with supplementation of different levels of oxidized fish oil on growth performance, nutrient digestibility, serum antioxidant activity and gut health in piglets. Forty-two piglets were arranged in 2 × 3 factorial treatments in a complete randomized block design with seven replicates per treatment and one pig per replicate for 28 d. Diets included twp types of maize (normal maize or aged maize) and three levels of oxidized fish oil (OFO) (3% non-oxidized fish oil (0% OFO), 1.5% OFO and 1.5% non-oxidized fish oil (1.5% OFO), and 3% OFO (3% OFO). Results showed that dietary aged maize did not affect growth performance, diarrhea, and the apparent total tract digestibility (ATTD) of nutrients in piglets (*p* > 0.05). However, aged maize increased malonaldehyde (MDA) content and decreased total antioxidant capacity (T-AOC) in serum on both 14th and 28th days (*p* < 0.05) compared to the normal maize groups. Meanwhile, compared with normal maize, dietary aged maize showed a slight, but not significant (*p* > 0.10) decrease in total superoxide dismutase (T-SOD) activity and VE content in serum on the 14th day. In addition, aged maize significantly decreased *GLUT2* mRNA expression (*p* < 0.05) and tended to increase (*p* < 0.10) *TNF-α* and *IL-6* mRNA expression in jejunal mucosa. Compared with non-oxidized fish oil, oxidized fish oil resulted in the decrease of the 14–28 d and 0–28 d ADG, as well as the ATTD of dry matter (DM), ether extract (EE), organic matter (OM) (*p* < 0.05), whereas the increase in diarrhea index (*p* < 0.05) and F/G of the whole period (*p* < 0.05). Oxidized fish oil decreased serum T-AOC on both the 14th and the 28th days (*p* < 0.05), and decreased serum T-SOD activity and VE content on the 28th day (*p* < 0.05), whereas increased serum MDA content on the 28th day (*p* < 0.05) and 14th day (*p* < 0.10) compared with fresh fish oil. Meanwhile, *MUC2* (*p* < 0.05) and *SGLT1* (*p* < 0.10) mRNA expression in jejunal mucosa were decreased compared with non-oxidized fish oil. In addition, dietary oxidized fish oil tended to decrease 14–28 d ADFI and the ATTD of CP (*p* < 0.10), and piglets fed oxidized fish oil significantly decreased 14–28 d ADFI, the ATTD of CP, *GLUT2* and *SGLT1* mRNA expressions in jejunal mucosa when piglet also fed with aged maize (*p* < 0.05). Collectively, these results indicated that dietary oxidized fish oil decreased growth performance and nutrients digestibility of piglets fed with aged maize. This nutrient interaction may be mediated by inhibiting intestinal nutrient transporter, inducing intestinal inflammation, and reducing antioxidant capacity.

## 1. Introduction

Maize is a primary source of energy and accounts for the highest proportion of ingredients in most livestock diets [1]. Large quantities of maize were stored for two years or more in grain depots to be prepared for the potential event of natural disasters. Unfortunately, a significant quantity of maize was lost due to biological, chemical and physical factors during these long storage times [2,3]. Storage time increases the contents of oxidation of lipid and fatty acid and also increases the fat acidity value (FAV) of maize [4,5]. Free fatty acids in maize are easily oxidized, producing lipid hydroperoxides, which affects catalase (CAT) and peroxidase (POD) activities [6]. The addition of dietary aged maize increases the amount of exogenous reactive oxygen species (ROS) and leads to lipid peroxidation, which may cause oxidative stress with negative impacts on animal health [7,8,9,10]. However, Mitchell and Beadles [11] reported that when maize was stored from 730 to 1020 d, there was no consistent or significant deterioration in the digestibility or the biological value of the maize when fed to rats. No effect was observed on the growth rate of rats fed maize which was stored for as long as six years [12]. Bartov [13] found that maize stored for 110 months in covered galvanized iron cans did not lead to any change in the chemical composition of the grains, except for a significant increase in lysine and a decrease in valine contents. Moreover, the nitrogen-corrected apparent metabolizable energy content for broiler chicks was not affected significantly by storage duration. These findings are not consistent with the negative effects of aged maize, that may lead to decreased growth performance with an increase in diarrhea in animals [14]. However, those previous studies utilized poultry and rats, and the effect of aged maize in piglets has not yet been studied. Most often, piglets are exposed to various complex stress, such as feed lipid oxidation or feed mildew [15]. To the best of our knowledge, there was no published reported the effects of feeding aged maize with or without oxidized oil in piglet diets, and whether the oxidized oil may exacerbate the effects of aged maize on piglets. Moreover, fish oil is rich in polyunsaturated fatty acids (PUFA) and which are easily oxidized [16]. Therefore, our objective was to evaluate the effects of aged maize on growth performance, nutrient digestibility, antioxidant activity and intestinal health of weaning piglets. Furthermore, we aimed to assess whether the effects of aged maize was exacerbated by adding oxidized fish oil at different levels.

## 2. Materials and Methods

All animal experiments were approved by the Animal Protection and Utilization Committee of Sichuan Agricultural University (Chengdu, China) [17]. The experiment described here was conducted at the Animal Experiment Center of Sichuan Agriculture University (Yaan, China).

### 2.1. Maize Samples

Aged maize and normal maize were obtained from National barns, China, after being stored for either five or 0.5 years. The maize was ground through a 1 mm screen using a hammer mill and then moisture (Method 930.15), lipids ether extracted (EE) (Method 920.39), crude ash assayed (Ash) (Method 942.05) and crude protein (CP) (Method 984.13) contents measured according to methods published by AOAC [18]. The amino acid content in maize was determined by automatic amino acid analyzer L-8900 (Japan Hitachi co., Ltd., Tokyo, Japan) according to Qiang, et al. [19]. Fatty acid content was determined using gas chromatography GC-17A (Shimadzu, Kyoto, Japan) according to Querijero, et al. [20]. Fatty acid value (FAV) was determined as potassium hydroxide to neutralize the acids in a100 gram of sample by method GB/T 20570-2015 [21]. The MDA content, CAT and POD activities were measured using assay kits (Jiancheng Bioengineering Institute of Biological Engineering, Nanjing, China). 

### 2.2. Animals and Diets

A total of forty-two 26 ± 1-d-old Duroc × Landrace × Yorkshire (DLY) cross-bred piglets (initial body weight 7.95 ± 0.56 kg) were allocated to six groups with seven piglets per group. The study was carried out as a 2 × 3 factorial experiment with two types of maize (normal maize or aged maize) and three levels of oxidized fish oil (OFO) (3% non-oxidized fish oil (0% OFO), 1.5% OFO and 1.5% non-oxidized fish oil (1.5% OFO), and 3% OFO (3% OFO)). Diets were formulated (Table 1) to meet the requirements of 10- to 20- kg pigs as established by the NRC (2012) [22]. All pigs were housed in individual metabolism cages (0.7 m× 1.5 m × 1 m) and were given *ad libitum* access to fresh water and feed. Room temperature was maintained at 25–28 °C, and relative humidity was maintained between 55–65%. Pigs were hand-fed three times per day (08:00, 14:00 and 20:00) to make sure that fresh feed was available.

### 2.3. Oxidized Fish Oil Preparation

Fresh fish oil was mixed with 143.54 mg/kg of FeSO_4_·7H_2_O, 58.94 mg/kg of CuSO_4_·5H_2_O, 600 mg/kg of H_2_O_2_ and 0.3% of H_2_O by volume, and heated to a constant temperature of 37 ± 1 °C. Throughout the heating process, air was continuously bubbled through the oil [23]. The degree of oxidation of the fish oil was monitored by the determination of the peroxide value (POV) every 24 h. 

After 13 days of oxidation, when the POV reached of 120.85 meq O_2_/kg, which was determined as the end point of oxidation. The POV of the fresh fish oil was 4.08 meq O_2_/kg. POV were determined according the method described by method GB 5009.227-2016 by potassium iodide titration [24]. The oxidized fish oil was then collected and stored at −20 °C until use. 

### 2.4. Sample Collection 

At the beginning of the trial, representative maize and feed samples of each group were sampled and stored at −20 °C for chemical analyses. Cr_2_O_3_ (0.3%) was included in all diets as an indigestible marker. After feeding diets with Cr_2_O_3_ from day 24 to day 27 of the experiment, fresh fecal samples were collected from all pigs. Each pig’s daily excreta was weighed, and 10 mL of a 10% H_2_SO_4_ solution was added to each 100 g of wet fecal sample and stored at −20 °C. At the end of this 4d period, all fecal samples of each pig were dried in an oven (65 °C for 72 h), ground up, and passed through a 1-mm sieve, then stored at −20 °C for later chemical analyses.

Piglets were fasted for 12-h prior to the morning of day 28, at which time blood samples were collected from the anterior juguldfar cava into vacuum tubes without anticoagulant. Blood samples were centrifuged at 3500× *g* for 15 min at 4 °C, and serum was separated and stored at −20 °C for further analyses. Pigs were then euthanized by intravenous injection of sodium pentobarbital (90 mg/kg BW). The abdomen was immediately opened and the jejunum was then opened longitudinally and washed with cold saline solution. Mucosal samples from mid- jejunum were dissected, snap-frozen in liquid nitrogen, and stored at −80 °C for further analyses. 

### 2.5. Mycotoxin Content of Maize and Diets

Normal maize, aged maize and diet samples were ground using a mill grinder (Retsch ZA 100; Restsch GmbH and Co., K. g., Haan, Germany) and passed through a 0.5-mm screen prior to analyses. Aflatoxin B1 (AFB1), zearalenone (ZEN), deoxynivalenol (DON), fumonisin (FB), ochratoxins (OTA) were measured using the national standard methods NY/T 2071-2011 [25], GB/T 30956-2014 [26], NY/T 1970-2010 [27] and GB/T 30957-2014 [28] by HPLC and a combination of gas chromatography or mass spectrometry methods [29]. 

### 2.6. Growth Performance

Body weights (BW) of pigs were individually measured after 12-h fasting on the mornings of day 0, day 14 and day 28 of the feeding trial, and feed intake per pen was collected daily throughout the trial to calculate average daily feed intake (ADFI), average daily gain (ADG) and ADFI/ADG (F/G).

### 2.7. Diarrhea Rate and Diarrhea Index

The pigs’ stool was observed at the same time of each morning and night by one observers blinded to treatments during the entire experimental period based on the method described by Yuan, et al. [30]. Briefly, fresh excreta from each pig was assessed visually with a score from 0 to 3 (0 = hard bar/hard granulous,1 = soft/forming, 2 = dense/not formed, 3 = fluid/not formed). The occurrence of diarrhea was defined as production of feces at level 2 or 3. Diarrhea frequency was then calculated as follows: Diarrhea rate (%) = (total number of pigs per pen with diarrhea)/(number of pigs per group × 28 d) × 100; Diarrhea index = SUM diarrhea score/(number of pigs per group × 28 d).

### 2.8. The Apparent Total Tract Digestibility

Diet and fecal samples were analyzed for DM, CP, EE, and Ash according to the procedures described by AOAC [18]. Moreover, organic matter (OM) was calculated by 100 minus the content of ash by AOAC [18]. The apparent total tract digestibility (ATTD) of DM, CP, EE, Ash, and OM were measured using Cr_2_O_3_ as an exogenous indicator. Concentration of Cr in the digested samples was measured using flame atomic absorption spectroscopy method AOAC [18]. The ATTD was calculated according to the following equation [31]: ATTD (%) = 100 − [(Cri/Cro) × (No/Ni)], in which Cri is the chromium concentration of diet; Cro is the chromium concentration of excreta; No is the nutrient concentration of excreta; Ni is the nutrient concentration in the diet.

### 2.9. Antioxidant Capacity in Serum

Serum total antioxidant capacity (T-AOC) with the colorimetric method, activities of total superoxide dismutase (T-SOD) with the hydroxylamine method, catalase (CAT) with the ammonium molybdate method and glutathione peroxidase (GSH-Px) with colorimetric method, contents of malondialdehyde (MDA) with the TBA method and vitamin E (VE) with the colorimetric method were measured using respective assay kits (Jiancheng Bioengineering Institute of Biological Engineering, Nanjing, China) with UV-VIS Spectrophotometer (UV1100, MAPADA, Shanghai, China) according to the manufacturer’s instructions.

### 2.10. Real-Time Quantitation PCR

Total RNA was extracted from jejunal mucosa using Trizol reagent (TaKaRa, Dalian, China) following the manufacturer’s instructions. The concentration and purity of RNA was analyzed spectrophotometrically (Beckman Coulter DU800; Beckman Coulter Inc., Pasadena, CA, USA). The integrity of RNA was checked by denaturing agarose gel electrophoresis. RNA samples were used to synthesize the cDNA using PrimeScripte RT reagent kit (Takara Shuzo Co., Ltd., Kusatsu, Japan) according to the manufacturer’s instructions. The cDNA was diluted and used as a PCR template to evaluate gene expression. The primers were synthesized commercially by Life Technologies Limited, and were listed in Table 2. Real-time PCR for quantification of Na+ glucose transporter 1 (*SGLT1*) , Glucose transporter 2 (*GLUT2*), Zonulaoccludens 1 (*ZO-1*), Occludin (*OCLN*), Claudin 1 (*CLDN1*), Mucin1 (*MUC1*), Mucin2 (*MUC2*), Tumor necrosis factor-α(*TNF-α*), Interleukin 1β (*IL-1β*), Interleukin 6 (*IL-6*), Interleukin 10 (*IL-10*) were performed on an Opticon DNA Engine (Bio-Rad) using SYBR Premix Ex TaqTM kits (TaKaRa) under the following conditions: pre-denaturation at 95 °C for 30 s and forty cycles of denaturation at 95 °C for 5 s, annealing at 60 °C for 30 s and extension at 72 °C for 60 s. A dissociation curve was constructed at the end of the reaction to ensure that only one amplication was formed. β-actin was chosen as the reference gene, and the relative expression ratio of the target genes in comparison with the reference gene was calculated as described by Pfaffl [32]. Mean Ct of each gene was determined from double measurements and normalized with the mean Ct of β-actin, a control gene.

### 2.11. Statistical Analyses

All data were analyzed as a 2 × 3 factorial with the general linear model for analysis of variance using SPSS (Version 22.0; IBM, Inc., Chicago, IL, USA). The model factors included the effects of maize (normal maize or aged maize), oxidized fish oil levels (0%, 1.5% or 3%), and their interaction. *p* < 0.05 was considered to indicate a significant difference and values between 0.05 and 0.10 to indicate a trend. Variable means for treatments showing significant differences in ANOVA were separated by Duncan’s test (*p* < 0.05). Data for diarrhea rate and diarrhea index were analyzed by using a non-parametric test. Values were expressed as means with their standard errors. 

## 3. Results

### 3.1. Physical and Chemical Properties of Aged Maize and Normal Maize

The data of physical and chemical properties of aged maize and normal maize are shown in Table 3. The moisture content of aged maize was slightly lower than that of normal maize, while the contents of EE, Ash and CP were slightly higher than that of normal maize. The composition of amino acids and fatty acids in aged maize are similar to those in normal maize. There were large differences in CAT and POD activities as well as in FAV and MDA content. Moreover, aged maize showed a greater decrease in CAT and POD activities and a greater increase in fat acid value and MDA content compared with normal maize. 

### 3.2. Mycotoxin Content of Maize and Diets

The level and type of mycotoxins in maize and diets are shown in Table 4. The contents of AFB1, ZEA, DON, FB, OTA in maize and diets were within GB 13078-2017 [33].

### 3.3. Growth Performance

The effects of dietary aged maize and oxidized fish oil levels on growth performance of piglets are shown in Table 5. Aged maize had no significant effect on BW, ADFI, ADG and F/G of piglets (*p* > 0.05). Dietary oxidized fish oil significantly decreased 14–28 d and 0–28 d ADG (*p* < 0.05), significantly increased 0–28 d F/G (*p* < 0.05), and tended to decrease 14–28 d ADFI of piglets (*p* < 0.10) compared to controls. There were no significant differences on the growth performance among the three normal maize groups (*p* > 0.05), while the 3% oxidized fish oil group significantly decreased 14–28 d ADFI and 0–28 d ADG, increased F/G of each stage compared to the 0% oxidized fish oil group of piglets fed with aged maize (*p* < 0.05). Maize × oil interaction showed a significant effect on 0–28 d F/G (*p* < 0.05).

### 3.4. Diarrhea Incidence

The effects of dietary aged maize and oxidized fish oil levels on diarrhea rate and diarrhea index of piglets are shown in Table 6. Aged maize tended to increase diarrhea index of piglets (*p* < 0.10). Dietary oxidized fish oil significantly increased diarrhea index of piglets (*p* < 0.05). No significant maize × oil interaction was observed on diarrhea rate and diarrhea index of piglets (*p* > 0.05).

### 3.5. Nutrient Digestibility

The effects of dietary aged maize and oxidized fish oil levels on nutrient digestibility of piglets are shown in Table 7. Aged maize had no significant effect on dietary nutrient digestibility of piglets (*p* > 0.05). Dietary oxidized fish oil significantly decreased the ATTD of DM, EE and OM (*p* < 0.05) and tended to decrease the ATTD of CP (*p* < 0.10) compared to controls. Moreover, the 3% oxidized fish oil group significantly decreased the ATTD of CP compared to the 0% oxidized fish oil group when piglets fed with aged maize (*p* < 0.05). A significant maize × oil interaction was observed on the ATTD of ash (*p* < 0.05).

### 3.6. Antioxidant Capacity in Serum

The effects of dietary aged maize and oxidized fish oil levels on antioxidant capacity in serum are shown in Table 8. Aged maize increased MDA concentration and decreased T-AOC activity in serum on both 14th and 28th days (*p* < 0.05), and showed a trend to decrease serum T-SOD activity and VE concentration on the 14th day (*p* < 0.10). Oxidized fish oil decreased T-AOC activity in serum on both the 14th and the 28th days (*p* < 0.05), and decreased serum T-SOD activity and VE concentration on the 28th day (*p* < 0.05), whereas increased serum MDA concentration on the 28th day (*p* < 0.05) and 14th day (*p* < 0.10) compared with fresh fish oil. No significant interaction between Maize × Oil was observed (*p* > 0.05).

### 3.7. Nutrient Transporter-Related Gene Expression in Jejunum Mucosa

The effects of dietary aged maize and oxidized fish oil levels on nutrient transporter-related gene expression in jejunum mucosa of piglets were shown in Table 9. Aged maize significantly decreased *GLUT2* mRNA expression in jejunal mucosa of piglets (*p* < 0.05) compared to controls. Dietary oxidized fish oil showed a trend to decrease *SGLT1* mRNA expression in jejunal mucosa of piglets (*p* < 0.10). The 3% oxidized fish oil group significantly decreased *GLUT2* and *SGLT1* mRNA expression in jejunal mucosa relative to 0% oxidized fish oil group of piglets fed with aged maize (*p* < 0.05). Maize × oil interaction effects showed no significant interactions on nutrient transporter-related gene expression in jejunum mucosa of piglets (*p* > 0.05).

### 3.8. Inflammation-Related Gene Expression in Jejunum Mucosa

The effects of dietary aged maize and oxidized fish oil levels on inflammation-related gene expression in jejunum mucosa of piglets were shown in Table 10. Aged maize showed a trend to increase *TNF-α* and *IL-6* mRNA expression in jejunal mucosa of piglets (*p* < 0.10). Dietary oxidized fish oil also showed a tend to increase *IL-6* mRNA expression in jejunal mucosa of piglets (*p* < 0.10). No significant maize × oil interaction effects were observed on inflammation-related gene expression in jejunum mucosa of piglets (*p* > 0.05).

### 3.9. Tight Junctions Protein and Mucins-Related Gene Expression in Jejunum Mucosa

The effects of dietary aged maize and oxidized fish oil levels on tight junction proteins and mucins-related gene expression in jejunal mucosa were shown in Table 11. Aged maize had no significantly effects on *ZO-1*, *OCLD*, *CLDN1*, *MUC1* and *MUC2* mRNA expression in jejunal mucosa of piglets (*p* > 0.05). Dietary oxidized fish oil significantly decreased *MUC2* mRNA expression in jejunal mucosa of piglets (*p* < 0.05) compared to controls. No significant interaction between maize × oil were observed on tight junction proteins and mucins-related gene expression in piglets (*p* > 0.05).

## 4. Discussion

In livestock production, numerous factors can induce oxidative stress and result in suboptimal health conditions of livestock and a reduction in production efficiency. A huge amount of maize used for feed production had been stored for a long time. Therefore, it is important to explore the effects of aged maize on piglets. Zhu, et al. [34] showed that 30% of aged maize was used to replace the normal maize significantly reduced ADG and final body weight of meat ducks. However, previous studies have reported that long-term storage of maize didn’t affect the growth of rats or broilers [12,13]. In order to further assess the effects of aged maize on the health of piglets, the present study was conducted to investigate an additional challenge with oxidized fish oil to determine if it would exacerbate the negative effect of aged maize on piglets. In our study, dietary aged maize had no significant effects on growth performance and nutrient digestibility of piglets, while dietary oxidized fish oil reduced growth performance and nutrient digestibility, and increased diarrhea index of piglets. This is consistent with previous results that oxidized fat significantly reduced the growth performance of piglets [23,35]. The reason may be that, on the one hand, lipid peroxidation products affect the quality of the diet and reduce its digestibility [36]. Alternatively, products of lipid peroxidation may cause certain pathological damage to the gastrointestinal tract of pigs, or may cause changes in cell membrane function [37,38,39]. Altered cell membrane function may in turn reduce feed digestibility in piglets leading to diarrhea and ultimately affect the normal growth and development. Further, we found that dietary oxidized fish oil tended to decrease 14–28 d ADFI of piglets, maize × oil interaction had a significant effect on 0–28 d F/G of piglets, in that the additional oxidized fish oil challenge exacerbated the effects of aged maize on growth performance of piglets.

To further explore the effects of aged maize and oxidized fish oil on absorption of nutrients, we investigated the relative mRNA expression of *SGLT1* and *GLUT2* following feeding of these nutrients to piglets. SGLT1, an apical intestinal transporter, is responsible for the majority of luminal glucose transport across intestinal epithelium, and is the rate-limiting step for absorption of dietary glucose [40,41]. GLUT2, a key transporter located at the basolateral membrane of the enterocyte, provides a glucose channel on enterocytes as a part of the intestinal glucose absorption system [42]. Therefore, the relative mRNA expression of *SGLT1* and *GLUT2* are closely related to the intestinal absorption of glucose. In this study, dietary aged maize significantly inhibited the relative mRNA expression of *GLUT2* and oxidized fish oil tended to decrease the relative mRNA expression of *SGLT1* in jejunal mucosa of piglets. Oxidized fish oil had no significant effect on the relative mRNA expression of *SGLT1* and *GLUT2* of piglets fed with normal maize, however the 3% oxidized fish oil group significantly decreased *GLUT2* and *SGLT1* mRNA expression in jejunal mucosa relative to 0% oxidized fish oil group of piglets fed with aged maize. These results suggested that aged maize had no significant effect on the nutrient transport under normal conditions and thus did not affect growth performance of piglets. However, piglets given an additional oxidative stress showed further reduced production compared to piglets fed normal maize. In the presence of oxidized fish oil, aged maize significantly reduced the relative mRNA expression of *GLUT2* and tended to reduce the relative mRNA expression of *SGLT1*. These observations indicate that the transport and absorption of glucose in the small intestine is negatively impacted by oxidized oil when additionally fed aged maize, that may lead to increased diarrhea and ultimately reducing the growth performance of piglets.

Reactive oxygen species (ROS) are always in an internal dynamic equilibrium state. When exogenous oxidative stress exceeds the resistance ability of the organism itself, which will reduce the antioxidant capacity of the organism and lead to oxidative damage [43]. MDA is a product of lipid peroxidation and its level is frequently used as a direct marker of lipid oxidative damage caused by ROS [44]. Antioxidant systems in organisms include enzymatic defense systems (CAT, GSH-Px, and SOD) and non-enzymatic defense systems (GSH, VE, and VC) that work together to eliminate excess free radicals to prevent oxidative damage [45,46]. Liu et al. [7] showed that broilers fed maize that had been stored for long periods of time had reduced serum GSH-Px and T-SOD activities and significantly increased serum MDA content compared to broiler-fed maize that had been stored for a short period of time. Our study supported those findings in that aged maize in piglet diet increased serum MDA content and decreased serum T-AOC, T-SOD activities and VE content compared to piglets fed maize stored for a short period of time. The reduction of antioxidant enzyme levels in aged maize and the accumulation of large amounts of ROS may have further reduced the antioxidant capacity of the diet. Dietary oxidized fish oil increased serum MDA content and reduced serum T-AOC activity and VE content in piglets. Studies by others found that dietary oxidized oil reduced the antioxidant capacity of pigs [47,48]. In our study, we also demonstrated that piglets fed aged maize + 3% oxidized fish oil group had the lowest antioxidant capacity of all treatment groups, which suggested that oxidized fish oil further decreased the antioxidant capacity when piglets fed with aged maize.

Oxidative stress is often accompanied by inflammation. Pro-inflammatory factors (such as *TNF-α*, *IL-1β*, *IL-6*) and anti-inflammatory factors (such as *IL-10*) that play crucial roles in the integrity of intestinal mucosal and tight junctional barriers [49]. Varady, et al. [50] found that oxidized fat markedly increased nuclear concentration of NF-κB in intestinal mucosa of mice. NF-κB plays a key role in regulating the expression of pro-inflammatory and anti-inflammatory factors in tissues. Manar, et al. [51] demonstrated that animals fed an oxidized diet showed enhanced plasma inflammatory markers (*IL-6* and *MCP-1*) and increased activation of NF-κB in the small intestine as well as decreased Paneth cell number compare to animals fed an unoxidized diet. These data suggested that the inflammatory response of the body was due to the oxidized oil in the diet which caused excessive ROS accumulation, thereby activating NF-κB, that in turn increased the expression of related pro-inflammatory factors. Similarly, in our study we found that feeding aged maize increased *TNF-α* and *IL-6* mRNA expression and the addition of oxidized fish oil in the diet increased *IL-6* mRNA expression in jejunal mucosa. All of which suggested that aged maize and oxidized fish oil elicited different degrees of jejunal inflammation in piglets. However, the specific inflammatory response induced by lipid peroxidation in aged maize and oxidized fish oil still needs further study.

Accumulating evidence has confirmed that inflammation is an important factor in intestinal barrier disruption [52]. Al and Boivin [53] found that cytokines could change the structure of tight junctions and modulate the expression of genes that code for tight junction proteins in mammals. Huang [54] showed that oxidized fish oil significantly decreased the relative expression of *Claudin-3*, *Claudin-15a*, *ZO-1*, *ZO-2* and *ZO-3* mRNAs in the intestinal mucosa of grass carp, while *OCLN* mRNA expression was also slightly decreased. In our current study, we found that aged maize and oxidized fish oil had no significant effect on the mRNA expression of *ZO-1, CLDN1* and *OCLN* in jejunal mucosa of piglets. Liu, et al. [55] reported that feeding weaned piglet diets that contained 10% thermally oxidized lipid for 38 d, appeared to impair oxidative status but had little influence on gut barrier, implying that pigs were relatively resilient to certain levels of lipid oxidation. The mucus layer is the first defense barrier against pathogen invasion in the gastrointestinal tract and an important part of the chemical barrier. Mucin proteins are a primary part of intestinal mucus that primarily includes MUC1 and MUC2 [56,57]. In our present study, dietary oxidized fish oil significantly reduced *MUC2* mRNA expression in the jejunal mucosa of piglets, which suggests that oxidized fish oil damages the intestinal chemical barrier of piglets.

## 5. Conclusions

In conclusion, the combination of aged maize with oxidized fish oil caused sufficient dietary stress that led to increased diarrhea and decreased growth performance due to reduce the antioxidant capacity, enhance the mRNA expressions of inflammatory factors and lead to inflammation in jejunal mucosa, inhibit the mRNA expressions of nutrient transporter and further reducing the apparent digestibility of nutrients. Either aged maize or oxidized fish oil alone did not cause sufficient digestive distress to produce diarrhea or impact growth performance.

## Figures and Tables

**Table 1 animals-09-00624-t001:** Composition and nutrient levels of the basal diet (air-dry basis %).

Items	Content	Nutrient Levels ^3^	Content
Maize	61.00	DE (Mcal/kg)	3.51
Soybean meal	17.71	CP	18.82
Fish meal	4.50	Ca	0.75
Whey powder (Low protein)	5.00	TP	0.56
Soybean protein concentrate	4.00	AP	0.37
Fish oil	3.00	Digestible Lys	1.30
Sucrose	1.00	Digestible Met + Cys	0.66
Glucose	1.50	Digestible Thr	0.77
Limestone	0.87	Digestible Trp	0.21
CaHPO_4_	0.36		
NaCl	0.20		
L-Lys · HCl (78%)	0.26		
DL-Met	0.06		
L-Thr (98.5%)	0.02		
Chloride choline (50%)	0.15		
Vitamin premix ^1^	0.03		
Mineral premix ^2^	0.30		
Total	100.00		

^1^ The vitamin premix provided the following per kg of the diet: VA 8000 IU, VD_3_ 2000 IU, VE 20 IU, VB_1_ 1.5 mg, VB_2_ 5.6 mg, VB_12_ 0.02 mg, VB_6_ 1.5 mg, D-D-Calcium Pantothenate 10 mg, Nicotinic acid 15 mg, Biotin 0.1 mg, Folic acid 0.6 mg. ^2^ The mineral premix provided the following per kg of the diet: Fe (as ferrous sulfate monohydrate)100 mg, Cu (as copper sulfate pentahydrate) 125 mg, Zn (as zinc sulfate) 100 mg, Mn (as manganese sulfate) 20 mg, I (as potassium iodide) 0.3 mg, Se (as sodium selenite) 0.3 mg. ^3^ Nutrient levels of diets were calculated by the values in Chinese feed database.

**Table 2 animals-09-00624-t002:** Primer sequences used for real-time PCR.

Gene	Primer Sequence (5′–3′)	Product Length(bp)	GeneBank Accession No.
*β-actin*	Forward: TCCATCGTCCACCGCAAATG	124	XM_003357928.4
	Reverse: TTCAGGAGGCTGGCATGAGG		
*SGLT1* ^1^	Forward: AGAAGGGCCCCAAAATGACC	96	NM_001164021.1
	Reverse: TGTTCACTACTGTCCGCCAC		
*GLUT2*	Forward: GACACGTTTTGGGTGTTCCG	156	NM_001097417.1
	Reverse: GAGGCTAGCAGATGCCGTAG		
*CLDN1*	Forward: TCTTAGTTGCCACAGCATGG	106	NM001244539
	Reverse: CCAGTGAAGAGAGCCTGACC		
*OCLN*	Forward: CTACTCGTCCAACGGGAAAG	158	NM_001163647.2
	Reverse: ACGCCTCCAAGTTACCACTG		
*ZO-1*	Forward: CAGCCCCCGTACATGGAGA	114	XM_005659811
	Reverse: GCGCAGACGGTGTTCATAGTT		
*MUC1*	Forward: GTGCCGCTGCCCACAACCTG	141	XM_001926883.4
	Reverse: AGCCGGGTACCCCAGACCCA		
*MUC2*	Forward: GGTCATGCTGGAGCTGGACAGT	181	XM_003122394.1
	Reverse: TGCCTCCTCGGGGTCGTCAC		
*IL-1β*	Forward: CAGCTGCAAATCTCTCACCA	112	NM_214055.1
	Reverse: TCTTCATCGGCTTCTCCACT		
*TNF-α*	Forward: CGTGAAGCTGAAAGACAACCAG	121	NM_214022.1
	Reverse: GATGGTGTGAGTGAGGAAAACG		
*IL-6*	Forward: TTCACCTCTCCGGACAAAAC	122	NM_001252429.1
	Reverse: TCTGCCAGTACCTCCTTGCT		
*IL-10*	Forward: TAATGCCGAAGGCAGAGAGT	134	NM_214041.1
	Reverse: GGCCTTGCTCTTGTTTTCAC		

^1^*SGLT1* = Na+ glucose transport protein-1; *GLUT2* = glucose transporter-2; *ZO-1* = Zonulaoccludens 1; *OCLN* = Occludin; *CLDN1* = Claudin 1; *MUC1* = Mucin1; *MUC2* = Mucin2, *TNF-α* = Tumor necrosis factor-α; *IL-1β* = Interleukin 1β; *IL-6* = Interleukin 6; *IL-10* = Interleukin 10.

**Table 3 animals-09-00624-t003:** Phytochemical properties of aged maize and normal maize (air dry).

Items	Normal Maize	Aged Maize
Moisture (%)	13.83	12.36
EE ^1^ (%)	3.77	4.67
Ash (%)	1.40	1.83
CP (%)	7.27	8.74
Amino acid (%)		
Glu	1.60	1.87
Gly	0.35	0.39
Ala	0.63	0.75
Cys	0.06	0.04
Val	0.39	0.42
Met	0.13	0.13
Ile	0.32	0.31
Leu	1.04	1.19
Tyr	0.29	0.34
Phe	0.46	0.46
Lys	0.30	0.26
His	0.24	0.26
Arg	0.36	0.35
Pro	0.72	0.83
Fatty acid (% of total fatty acid)		
Myristic acid (C16:0)	12.96	12.67
Palmitic acid (C18:0)	1.56	1.75
Arachidic acid (C20:0)	0.37	0.35
Oleic acid (C18:1n-9)	26.78	24.48
Eicosenoic acid (C20:1n-9)	0.30	0.24
Linoleic acid (C18:2n-6)	55.98	58.36
α-Linolenic acid (C18:3n-3)	1.50	0.99
FAV (mg KOH/100g)	65.37	128.25
MDA (nmol/mg prot)	8.31	22.37
POD (U/mg prot)	15.19	3.05
CAT (U/mg prot)	8.53	1.17

^1^ EE = ether extract; Ash = crude ash; CP = crude protein; FAV = fatty acid value; MDA = malondialdehyde; POD = peroxidase; CAT = catalase.

**Table 4 animals-09-00624-t004:** Mycotoxin content of maize and diets (μg/kg).

Items	Normal Maize	Aged Maize	Limits in Maize ^4^	Normal Maize Diets			Aged Maize Diets			Limits in Diets
				0 ^1^	1.5	3	0	1.5	3	
AFB1 ^2^	ND ^3^	ND	50	ND	ND	ND	ND	ND	ND	10
ZEA	ND	18.7	500	27.5	26.7	23.6	44.7	35.1	40.1	150
DON	ND	400	5000	ND	ND	ND	400	300	300	1000
FB	263	ND	6000	422	511	674	83	73	81	5000
OTA	ND	ND	100	ND	ND	ND	ND	ND	ND	100

^1^ 0, including 3% fresh fish oil and 0% oxidized fish oil; 1.5, including 1.5% fresh fish oil and 1.5% oxidized fish oil; 3, including 0% fresh fish oil and 3% oxidized fish oil. ^2^ AFB1, aflatoxin B1; ZEA, zearalenone; DON, vomitoxin; FB, fumonisin; OTA, ochratoxin. ^3^ ND: not detected. Detection limit of AFB1 was 1.0 μg/kg; Detection limit of ZEA was 5.0 μg/kg; Detection limit of DON was 100 μg/kg; Detection limit of FB was 50 μg/kg; Detection limit of OTA was 5.0 μg/kg. ^4^ The data of limits on mycotoxins in maize and feed were from China Feed Hygiene Standard (GB 13078-2017).

**Table 5 animals-09-00624-t005:** Effects of aged maize and oxidized fish oil levels on growth performance of weaning piglets.

Items	Normal Maize			Aged Maize			SEM	*p*-Value		
	0 ^1^	1.5	3	0	1.5	3		Maize ^2^	Oil	Maize × Oil
Initial BW ^3^ (kg)	7.91	7.95	7.99	7.95	7.95	7.95	0.09	0.998	0.984	0.982
14 d BW (kg)	11.61	11.56	11.43	11.86	11.37	11.17	0.20	0.882	0.686	0.863
28 d BW (kg)	17.59	17.34	16.94	18.06	16.98	16.10	0.28	0.612	0.227	0.703
Phase 1 0–14 d										
ADFI (g)	437.76	440.41	409.64	445.51	397.55	417.55	14.38	0.767	0.725	0.731
ADG (g)	264.69	257.96	245.48	279.59	244.90	230.61	10.06	0.836	0.417	0.807
F/G	1.68 ^ab^	1.74 ^ab^	1.67 ^ab^	1.60 ^b^	1.64 ^ab^	1.84 ^a^	0.03	0.950	0.284	0.124
Phase 2 14–28 d										
ADFI (g)	747.86 ^a^	740.20 ^a^	699.64 ^ab^	741.53 ^a^	707.76 ^ab^	633.57 ^b^	14.05	0.207	0.073	0.678
ADG (g)	426.73 ^a^	412.76 ^ab^	393.93 ^ab^	442.45 ^a^	400.31 ^ab^	351.84 ^b^	9.67	0.483	0.033	0.447
F/G	1.76 ^ab^	1.79 ^ab^	1.79 ^ab^	1.69 ^b^	1.77 ^ab^	1.90 ^a^	0.02	0.614	0.102	0.250
Overall 0–28 d										
ADFI (g)	592.81	590.31	554.64	593.52	552.70	516.67	12.97	0.347	0.220	0.782
ADG (g)	345.71 ^a^	335.36 ^ab^	319.70 ^ab^	355.92 ^a^	322.60 ^ab^	275.48 ^b^	8.61	0.347	0.043	0.415
F/G	1.72 ^b^	1.77 ^b^	1.73 ^b^	1.67 ^b^	1.72 ^b^	1.89 ^a^	0.02	0.540	0.026	0.024

^1^ 0, including 3% fresh fish oil and 0% oxidized fish oil; 1.5, including 1.5% fresh fish oil and 1.5% oxidized fish oil; 3, including 0% fresh fish oil and 3% oxidized fish oil. ^2^ Maize: normal maize or aged maize; Oil: 0%, 1.5% or 3% oxidized fish oil; maize × oil: maize × oxidized fish oil levels. ^3^ BW, body weight; ADFI, average daily feed intake; ADG, average daily gain; F/G, feed/gain. ^a, b^ In the same row, values with different letter superscripts mean significant difference (*p* < 0.05).

**Table 6 animals-09-00624-t006:** Effects of aged maize and oxidized fish oil levels on diarrhea rate and diarrhea index of weaning piglets.

Items	Normal Maize			Aged Maize			SEM	*p*-Value		
	0 ^1^	1.5	3	0	1.5	3		Maize ^2^	Oil	Maize × Oil
0–28 d										
Diarrhea rate (%)	0.51	1.02	3.06	1.02	3.57	2.55	0.63	0.318	0.296	0.842
Diarrhea index	0.01	0.03	0.12	0.02	0.09	0.17	0.02	0.089	0.030	0.352

^1^ 0, including 3% fresh fish oil and 0% oxidized fish oil; 1.5, including 1.5% fresh fish oil and 1.5% oxidized fish oil; 3, including 0% fresh fish oil and 3% oxidized fish oil. ^2^ Maize: normal maize or aged maize; oil: 0%, 1.5% or 3% oxidized fish oil; maize × oil: maize × oxidized fish oil levels.

**Table 7 animals-09-00624-t007:** Effects of aged maize and oxidized fish oil levels on nutrients apparent digestibility of weaning piglets.

Items	Normal Maize			Aged Maize			SEM	*p*-Value		
	0 ^1^	1.5	3	0	1.5	3		Maize ^2^	Oil	Maize × Oil
28 d										
DM ^3^ (%)	82.96 ^a^	82.25 ^ab^	82.00 ^ab^	83.49 ^a^	81.38 ^ab^	79.84 ^b^	0.35	0.207	0.023	0.257
EE (%)	78.16 ^a^	75.39 ^ab^	72.27 ^cd^	77.62 ^ab^	74.81 ^bc^	71.17 ^d^	0.55	0.379	<0.001	0.953
Ash (%)	51.94 ^bc^	52.01 ^bc^	54.10 ^ab^	57.92 ^a^	54.22 ^ab^	47.90 ^c^	0.77	0.619	0.066	0.002
OM (%)	85.02 ^a^	83.91 ^ab^	83.56 ^ab^	84.98 ^a^	82.95 ^ab^	81.73 ^b^	0.34	0.141	0.013	0.516
CP (%)	77.38 ^ab^	76.07 ^ab^	76.23 ^ab^	78.49 ^a^	75.68 ^ab^	73.29 ^b^	0.60	0.533	0.096	0.384

^1^ 0, including 3% fresh fish oil and 0% oxidized fish oil; 1.5, including 1.5% fresh fish oil and 1.5% oxidized fish oil; 3, including 0% fresh fish oil and 3% oxidized fish oil. ^2^ Maize: normal maize or aged maize; Oil: 0%, 1.5% or 3% oxidized fish oil; maize × oil: maize × oxidized fish oil levels. ^3^ DM, dry matter; EE, crude fat; ash, crude ash; OM, organic matter; CP, crude protein. ^a, b, c^ In the same row, values with different letter superscripts mean significant difference (*p* < 0.05).

**Table 8 animals-09-00624-t008:** Effects of aged maize and oxidized fish oil levels on antioxidant capacity in serum of weaning piglets.

Items	Normal Maize			Aged Maize			SEM	*p*-Value		
	0 ^1^	1.5	3	0	1.5	3		Maize ^2^	Oil	Maize × Oil
14 d										
MDA ^3^ (nmol/mL)	5.81 ^b^	6.23 ^ab^	6.51 ^ab^	6.49 ^ab^	6.78 ^ab^	7.18 ^a^	0.14	0.017	0.098	0.971
T-AOC (U/mL)	1.30 ^a^	1.18 ^a^	0.83 ^ab^	1.06 ^ab^	0.88 ^ab^	0.62 ^b^	0.06	0.029	0.005	0.956
T-SOD (U/mL)	148.69	148.57	143.64	138.26	134.77	137.06	2.76	0.075	0.901	0.871
GSH-Px (U/mL)	539.82	520.24	519.03	529.24	520.41	517.98	10.94	0.870	0.827	0.979
CAT (U/mL)	10.84	10.01	9.49	9.99	9.35	8.26	0.45	0.325	0.399	0.968
VE (μg/mL)	6.73 ^a^	6.20 ^ab^	5.62 ^ab^	6.24 ^ab^	5.19 ^ab^	4.67 ^b^	0.24	0.088	0.077	0.887
28 d										
MDA (nmol/mL)	4.04 ^b^	4.32 ^b^	4.89 ^ab^	4.46 ^b^	5.06 ^ab^	6.37 ^a^	0.23	0.041	0.032	0.578
T-AOC (U/mL)	1.24 ^a^	1.02 ^ab^	0.90 ^bc^	0.89 ^bc^	0.79 ^bc^	0.69 ^c^	0.05	0.003	0.033	0.752
T-SOD (U/mL)	128.90 ^ab^	128.01 ^ab^	123.95 ^ab^	131.07 ^a^	121.50 ^b^	120.98 ^b^	1.15	0.255	0.019	0.262
GSH-Px (U/mL)	547.60	540.38	530.37	508.41	508.12	503.41	10.44	0.136	0.912	0.973
CAT (U/mL)	12.78	12.08	11.77	12.09	11.82	11.20	0.52	0.646	0.783	0.986
VE (μg/mL)	5.51 ^a^	4.99 ^ab^	4.38 ^b^	4.93 ^ab^	4.88 ^ab^	4.08 ^b^	0.14	0.220	0.012	0.757

^1^ 0, including 3% fresh fish oil and 0% oxidized fish oil; 1.5, including 1.5% fresh fish oil and 1.5% oxidized fish oil; 3, including 0% fresh fish oil and 3% oxidized fish oil. ^2^ Maize: normal maize or aged maize; Oil: 0%, 1.5% or 3% oxidized fish oil; Maize × Oil: maize × oxidized fish oil levels. ^3^ MDA, malondialdehyde; T-AOC, total antioxidant capacity; T-SOD, total superoxide dismutase; GSH-Px, glutathione peroxidase; CAT, catalase; VE, vitamin E. ^a, b, c^ In the same row, values with different letter superscripts mean significant difference (*p* < 0.05).

**Table 9 animals-09-00624-t009:** Effects of aged maize and oxidized fish oil levels on nutrient transporter-related gene expression in jejunum mucosa of weaning piglets.

Items	Normal Maize			Aged Maize			SEM	*p*-Value		
	0 ^1^	1.5	3	0	1.5	3		Maize ^2^	Oil	Maize × Oil
28 d										
*SGLT1* ^3^	1.00 ^ab^	0.97 ^ab^	0.77 ^ab^	1.40 ^a^	0.79 ^ab^	0.56 ^b^	0.09	0.980	0.056	0.284
*GLUT2*	1.00 ^a^	0.89 ^ab^	1.03 ^a^	0.94 ^a^	0.72 ^ab^	0.42 ^b^	0.07	0.041	0.300	0.230

^1^ 0, including 3% fresh fish oil and 0% oxidized fish oil; 1.5, including 1.5% fresh fish oil and 1.5% oxidized fish oil; 3, including 0% fresh fish oil and 3% oxidized fish oil. ^2^ Maize: normal maize or aged maize; oil: 0%, 1.5% or 3% oxidized fish oil; maize × oil: maize × oxidized fish oil levels. ^3^ SGLT1, Na+ dependent glucose transporter; GLUT2, glucose transporter 2; ^a, b^ In the same row, values with different letter superscripts mean significant difference (*p* < 0.05).

**Table 10 animals-09-00624-t010:** Effects of aged maize and oxidized fish oil levels on inflammation-related gene expression in jejunum mucosa of weaning piglets.

Items	Normal Maize			Aged Maize			SEM	*p*-Value		
	0 ^1^	1.5	3	0	1.5	3		Maize ^2^	Oil	Maize × Oil
28 d										
*TNF-α* ^3^	1.00	1.35	1.52	1.51	1.75	1.82	0.10	0.059	0.268	0.917
*IL-1β*	1.00	1.11	1.29	1.24	1.35	1.69	0.12	0.238	0.442	0.953
*IL-6*	1.00 ^b^	1.18 ^ab^	1.64 ^ab^	1.46 ^ab^	1.59 ^ab^	1.93 ^a^	0.10	0.060	0.076	0.936
*IL-10*	1.00	0.89	0.72	0.82	0.54	0.44	0.09	0.167	0.377	0.927

^1^ 0, including 3% fresh fish oil and 0% oxidized fish oil; 1.5, including 1.5% fresh fish oil and 1.5% oxidized fish oil; 3, including 0% fresh fish oil and 3% oxidized fish oil. ^2^ Maize: normal maize or aged maize; oil: 0%, 1.5% or 3% oxidized fish oil; maize × oil: maize × oxidized fish oil levels. ^3^
*TNF-α*, Tumor necrosis factor-α; *IL-1β*, Interleukin 1β; *IL-6*, Interleukin 6; *IL-10*, Interleukin 10. ^a, b^ In the same row, values with different letter superscripts mean significant difference (*p* < 0.05).

**Table 11 animals-09-00624-t011:** Effects of aged maize and oxidized fish oil levels on tight junctions protein and mucins-related gene expression in jejunum mucosa of weaning piglets.

Items	Normal Maize			Aged Maize			SEM	*p*-Value		
	0 ^1^	1.5	3	0	1.5	3		Maize ^2^	Oil	Maize × Oil
28 d										
*ZO-1* ^3^	1.00	0.64	0.77	1.05	0.80	0.71	0.08	0.782	0.288	0.886
*OCLN*	1.00	1.03	0.92	0.91	0.98	0.95	0.05	0.736	0.872	0.905
*CLDN1*	1.00	1.14	0.83	1.01	1.04	0.95	0.05	0.932	0.301	0.701
*MUC1*	1.00	0.86	0.85	1.12	0.82	0.72	0.10	0.922	0.429	0.844
*MUC2*	1.00 ^a^	0.70 ^ab^	0.69 ^ab^	0.87 ^ab^	0.73 ^ab^	0.52 ^b^	0.05	0.378	0.043	0.707

^1^ 0, including 3% fresh fish oil and 0% oxidized fish oil; 1.5, including 1.5% fresh fish oil and 1.5% oxidized fish oil; 3, including 0% fresh fish oil and 3% oxidized fish oil. ^2^ Maize: normal maize or aged maize; oil: 0%, 1.5% or 3% oxidized fish oil; maize × oil: maize × oxidized fish oil levels. ^3^
*ZO-1*, Zonulaoccludens 1; *OCLN*, Occludin; *CLDN1*, Claudin 1; *MUC1*, Mucin1; *MUC2*, Mucin2. ^a, b^ In the same row, values with different letter superscripts mean significant difference (*p* < 0.05).

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
