# Peer review of "Effects of Dietary Aged Maize with Oxidized Fish Oil on Growth Performance, Antioxidant Capacity and Intestinal Health in Weaned Piglets"

_animals, 2019, doi:10.3390/ani9090624_

Round 1

Reviewer 1 Report

Piglets fed aged maize with oxidized fish oil have increased diarrhea and reduced growth performance due to effects on antioxidant capacity and intestinal health

General comments:

In the article entitled “Piglets fed aged maize with oxidized fish oil have increased diarrhea and reduced growth performance due to effects on antioxidant capacity and intestinal health” authors investigated the effect of “normal” and “aged” maize together with inclusion of different levels of oxidized fish oil on growth performance, parameters of antioxidative defense in serum (MDA, vitamin E and antioxidant enzymes) and on intestinal health (relative expression of glucose transport genes, relative expression of inflammation-related genes and relative expression of tight junction proteins and mucins) of piglets. Many countries have large quantities of grain reserves in case of emergencies, which have to be replaced regularly.  Replaced grains can then be used as feed components for animals; therefore, the topic of the article is, in my opinion, interesting for the research community and have practical implications.

The experimental design is, in my opinion, appropriate, consisting of two maize groups (controls) and two levels of oxidized fish oil. Selected nutrient apparent digestibilities were measured using Cr2O3 marker technique, the selection of analytical parameters of antioxidative defense and intestinal health is wide and proper

Main issues to be addressed and corrected before the article is suitable for publication:

The title of the article should be in my opinion rewritten, as diarrhea scores and growth performance was assessed during a relatively short period (28 days), using young animals, on a small number of animals, so the results should be interpreted with caution, this type of conclusions in the title would require large-scale experiment. Term “antioxidant capacity” also do not reflect all parameters that were analyzed, consider revision. Simple abstract, abstract and conclusions: the same as above, for diarrhea scores and growth performance draw the conclusions in all sections of the article with caution. In my opinion, other outcomes of the research are more important, as stored maize was used in large proportion in the diet of young animals without a major negative effect on gut health Introduction: why fresh/oxidized fish oil was used in the diets? Is it a common ingredient in feed mixtures for piglets? Moreover, why fish oil was oxidized? Please, explain and add information in the introduction and discussion where appropriate. Please, clearly explain and discuss, the reasoning behind testing both, differently stored maize and different levels of oxidized fish oil in the same experiment? Material and methods/animals and diets: please, clearly indicate that fresh fish oil was partially/totally replaced with oxidized oil. Feed analysis data is missing, only calculated nutrient levels are reported, please consider adding analytical results to the table 1. Phases of the experiment should be described, denoted in tables 5 and 8, please add where appropriate. Material and methods/oxidized fish oil preparation: Fenton reaction was used to oxidize oil at low temperatures. How much time was needed to obtain oil with POV >100? POV was monitored by a standard method, which is not available to readers. Please, add a brief description of the method. As lipid peroxides are decomposed to aldehydes and ketones, especially in the presence of iron and copper, have you checked POV before mixing oxidized oil into feeds (after storage)? Results: Diarrhea incidence and rate: Incidence was very low (1-day diarrhea in fresh maize group, up to 7 days in aged maize+1.5 oxidized FO group) and SEM is quite high (0.63 in diarrhea rate, 0.02 in diarrhea index compared to average rates and indexes), were the data normally distributed? If not, Duncan test is in my opinion not correct test for statistical evaluation of the obtained data. Please, verify the statistical significance of results using more appropriate statistical methods. Discussion: should be improved, the firm structure is needed. Discussion on nutrient composition of “normal” and “aged” maize is missing, discussion on the effect of aged maize and levels of oxidized fish oil on nutrients apparent digestibility should be addressed. Products of lipid oxidation in this experiment have two possible sources (oxidized lipids in maize – indicated with high secondary products of oxidation, MDA value in stored maize and in oxidized fish oil, indicated with high primary products of oxidation - POV), which needs to be addressed.

Specific comments:

Simple Summary:

Line 17: “aged maize”, the term “aged maize” in my opinion would implicate accelerated aging, consider revision (e.g. stored maize) Lines 23-24: »inhibited the expression of intestinal nutrient transport«, did you mean expression of genes, please correct.

Abstract:

Line 36: »normal maize « a bit awkward term, which would imply that “aged maize” is not normal, please consider revision (e.g. fresh).

Introduction

Line 63: “FFA are easily oxidized, producing H2O2”. You probably mean, producing lipid hydroperoxides. Please, explain and/or rephrase

Materials and methods:

Line 92: Fatty acid value – you probably mead acid value, which is expressed as mg KOH/mass of the sample, please correct also in table 3 As maize contains oils, prone to oxidation, was POV and MDA analyzed in maize, oil and feed samples? If it was, please include the values in tables 1 and 3 and discuss the obtained levels in Discussion. Line 154-155: numbers of pigs per pen – you probably mean the number of pigs per group? Line 159: organic matter was not analyzed, it was calculated Line 168: the content of MDA was analyzed using kit. Please add a brief description of the method. Is the method specific to MDA or not (TBARS)? Line 169: vitamin E was measured using kits – can you please add information, which isomers are being analyzed (only a-tocopherol or all isomers), is the method specific for MDA or are there any known interferences?

Results:

Line 202: “physical and chemical characteristics” – please, explain, which physical characteristics are reported, or rephrase Line 273: “... on antioxidant activity” – you probably mean all analyzed parameters of antioxidant defense system, the same in the title of table 8, please rephrase Table 8: vitamin E levels in serum are quite high considering the level of dietary supplementation (20 IU) and the increased load of pro-oxidants. Do you have an explanation for such a result? Table 8, title: please rephrase the title Table 10, title: “inflammation-related gene”, please change to inflammation-related gene expression Table 11, title: correct font size, footnote: correct superscript in a,b,c

Discussion

Lines 373: “additional oxidative stress showed further reduced production” of what? Please, explain Lines 373-374: “in the presence of oxidized fish oil, aged maize significantly reduced the relative mRNA…” – if I understand results in table 9 correctly, there was only tendency ( P=0.056), please explain and/or rephrase Line 376 “oxitative oil”, probably you mean oxidized oil

Author Response

Reviewer 1

Response: In the manuscript, we have made some corrections as follows:

1) In the title, “Piglets fed aged maize with oxidized fish oil have increased diarrhea and reduced growth performance due to the effects on antioxidant capacity and intestinal health” was corrected as “Effects of dietary aged maize with oxidized fish oil on growth performance, antioxidant capacity and intestinal health in weaned piglets”.

2) Simple abstract, abstract and conclusions: In the simple abstract, abstract and conclusions: Under normal conditions, aged maize had no significant effect on growth performance, diarrhea and nutrient digestibility of the piglets, but reduced serum antioxidant capacity. However, when oxidized fish oil was added, aged maize reduced serum antioxidant capacity further, inhibited the expressions of genes related to intestinal nutrient transport, promoted intestinal inflammation, and also reduced the ATTD of nutrients, increased diarrhea and finally reduced the growth performance of piglets.

3) Introduction: Most often, piglets are exposed to various complex stress, such as lipid oxidation and feed mildew [15]. To the best of our knowledge, there was no published reported effects of feeding aging maize with or without oxidized oil in piglet diets and whether the oxidized oil may increase the effects of aging maize on piglets. Moreover, fish oil is rich in polyunsaturated fatty acids (PUFA) which are easily oxidized [16].

4) Material and methods

The study was carried out as a 2 × 3 factorial experiment with 2 types maize (normal maize or aged maize) and 3 levels of oxidized fish oil (OFO) (3% non-oxidized fish oil (0% OFO), 1.5% OFO and 1.5% non-oxidized fish oil (1.5% OFO) , and 3% oxidized fish oil (3% OFO) ). Thank you for your suggestion, but for various reasons, it is impossible to measure again now. Added phases of the experiment to the table.

5) Material and methods

After 13 days of oxidation, when the POV reached of 120.85 meq O2/kg, which was determined as the end point of oxidation. The POV of the fresh fish oil was 4.08 meq O2/kg. POV were determined according the method described by method GB 5009.227-2016 by potassium iodide titration. Oxidized fish oil was stored in the refrigerator immediately after preparation for storage, and the feed was prepared within two days, so the POV before mixing was not determined.

6) Results: Thank you for your suggestion. According to the statistical analysis teacher's suggestion, data for diarrhea rate and diarrhea index were analyzed by using a non-parametric test.

7) Discussion:

Thank you for your suggestion. We all know that the nutrient content of each batch of maize will vary greatly, even if it is produced in the same variety. The purpose of this paper is to investigate the effects of different storage years of maize on growth and intestinal health in piglets. Therefore, it is not meaningful to discuss the difference in nutrient between the two maize, their biggest difference is the difference in some biochemical indicators caused by storage time, and has been given. Thank you for your suggestion. Make some modifications in the paper. Thank you for your suggestion. In China, the freshness of corn is often judged by the fatty acid value and MDA content, and the oil is often judged by acid value and POV, so I think this is reasonable.

Specific comments:

Simple Summary:

Line 17: Thank you for your suggestion. We think aged maize is reasonable.

Lines 23-24: “inhibited the expressions of intestinal nutrient transport” was corrected as “inhibited the expressions of genes related to intestinal nutrient transport”

Abstract:

Line 36: Thank you for your suggestion, but it is true that many indicators of aging corn have changed compared to normal corn, so it can be distinguished in this way.

Introduction:

Line 63: “producing H2O2” was corrected as “producing lipid hydroperoxides”.

Materials and methods:

Line 92: Thank you for your suggestion. In China, maize has different determine indicators, such as fatty acid value, MDA content and POD and CAT enzyme activity.

Line 154-155: “number of pigs per pen” was corrected as “number of pigs per group”.

Line 159: “organic matter (OM) was analyzed by” was corrected as “organic matter (OM) was calculated by”

Line 168: content of malondialdehyde (MDA) TBA method.

Line 169: content of vitamin E (VE) with colorimetric method.

Results:

Line 202: “physical and chemical properties” was corrected as “Phytochemical properties”

Line 273: “on antioxidant activity” was corrected as “on antioxidant capacity”

Thank you for your suggestion. Vitamin E levels in serum in a reasonable interval.

Table 10, “inflammation-related gene” was corrected as “inflammation-related gene expression”

title: has been modified

footnote: has been modified

Discussion:

Lines 373: “additional oxidative stress showed further reduced production”: The study found that aged maize had little effect on growth performance, diarrhea, nutrient digestibility, intestinal health related indicators under normal oil, while in the premise of oxidized oil, aged maize significantly reduced growth performance and affected intestinal health compared with normal maize.

Lines 373-374: “in the presence of oxidized fish oil, aged maize significantly reduced the relative mRNA…” was corrected as “In the presence of oxidized fish oil, aged maize significantly reduced the relative mRNA expression of GLUT2 and tended to reduce the relative mRNA expression of SGLT1”.

Line 376: “oxidative oil” was corrected as “oxidized oil”

Sincerely,

Bing Yu

Reviewer 2 Report

The case of implementation of  corn (stored for log period of time) is very interesting from practical point of view.

General idea about action of oxidazed nutrients on piglets is well known however this paper is presenting results of many different estimations using modern methodology and style of this presentation is OK. For me the addition of oxidized fish oil is a bit artificial.

Methods section concerning antioxidants needs addition of methodology. It is not enough to say that you used assay kit. E.g. when you discribing estimation of GSH-Px in serum - is this GSH-Px 2? Which method as used?

Line 274 - T-AOC capacity, vit E concentration, enzymes activity

You are giving information that as a final product of FA oxidation H2O2 is formed - please give citation.

Line 98 animals ar not replicants

102 - free acces or ad libitum

103 - controled kept ?

Please give citation concerning method of fat oxidation 

129- Insted of screen use sive

150 - I dont understand sentence: observers nat aware of the treatment

159-               -//-             OM was analyzed by 100 minus ... ash by AOAC.

161 - Cr in digested samples.  Stols?

344 - ... further understand the ...... ?

Author Response

Dear editors and reviewers,

We would like to thank you for giving us an opportunity to revise our manuscript. We also appreciate very much for your positive and constructive comments and suggestions on our manuscript entitled “Effects of dietary aged maize with oxidized fish oil on growth performance, antioxidant capacity and intestinal health in weaned piglets”. Those comments are all valuable and very helpful for revising and improving our paper. We have studied comments carefully and have made correction. All amendments are highlighted in red in the revised manuscript.. The point-by-point response to the comments and suggestions are listed as below. We hope the new manuscript will meet the requirements. Looking forward to hearing from you!

Thank you and best regards.

Yours sincerely,

Bing Yu

Corresponding author,

Name: Bing Yu

E-mail: ybingtian@163.com

Responds to the reviewer’s comments:

Reviewer 2

For me the addition of oxidized fish oil is a bit artificial. Piglets will face more oxidative stress in actual production, and this experiment is very meaningful to add oxidized fish oil to aged maize to further explore the effects of aged maize on piglet health. Methods section concerning antioxidants needs addition of methodology. It is not enough to say that you used assay kit. E.g. was corrected as “Serum total antioxidant capacity (T-AOC) with colorimetric method, activities of total superoxide dismutase (T-SOD) with hydroxylamine method, catalase (CAT) with ammonium molybdate method and glutathione peroxidase (GSH-Px) with colorimetric method, contents of malondialdehyde (MDA) TBA method and vitamin E (VE) with colorimetric method were measured using respective assay kits (Jiancheng Bioengineering Institute of Biological Engineering, Nanjing, China) with UV-VIS Spectrophotometer (UV1100, MAPADA, Shanghai, China) according to the manufacturer’s instructions”. “Line 274 - T-AOC capacity, vit E concentration, enzymes activity” was corrected as “antioxidant capacity, T-AOC activity, VE concentration, T-SOD activity, MDA concentration”. You are giving information that as a final product of FA oxidation H2O2 is formed - please give citation. Thank you for your suggestion. Reference to Rios-Gonzalez (2002). Line 98 animals are not replicants. was corrected as “six groups with seven pigs per group”. “Line 102 - free access or ad libitum”. Thank you for your suggestion. I think is ad libitum. “Line 103 - controled kept ?” was corrected as “maintained”. “Please give citation concerning method of fat oxidation”. Thank you for your suggestion. Reference to Yuan et al. (2007) for the method of oil oxidation. “Line 129- Instead of screen use sive” was corrected as “1-mm sieve” “Line 150 - I dont understand sentence: observers nat aware of the treatment” was corrected as “The pigs’ stool was observed at the same time of each morning and night by one observers blinded to treatments in the entire experimental period based on the method described by Yuan, et al. [27]” “Line 159- OM was analyzed by 100 minus ... ash by AOAC” was corrected as “organic matter (OM) was calculated by” “Line 161 - Cr in digested samples. Stols?”. Thank you for your suggestion, digested samples means samples that have been digested, including feed and feces. “Line 344 - ... further understand the ......?” was corrected as “further assess the”.

Sincerely,

Bing Yu

Round 2

Reviewer 1 Report

As the authors addressed all previous comments and suggestions, I recommend for the article to be accepted.

  Best regards.